# Direct aluminium-alloy upcycling from entire end-of life vehicles

Patrick Krall ®[1], Irmgard Weißensteiner ®[2], Philip Aster ®[2],
Phillip Dumitraschkewitz ®[1], Matheus A. Tunes ®[1], Thomas Kremmer ®[1],
Sebastian Samberger ®[1], Bernhard Trink[3] & Stefan Pogatscher ®[1] ✉

The global transition to a circular economy hinges on the development of sustainable recycling processes for end-of-life vehicles. Ongoing electrification and material choices over the recent decades hinder their integration in existing recycling pathways. This results in a large surplus of low-grade aluminium scraps and forfeits substantial energy, emissions, and cost savings, making the need for novel recycling approaches an urgent problem. This study presents a process for directly upcycling mixed end-of-life vehicles scrap into a high-performance aluminium alloy under realistic industrial conditions. It is compatible with existing infrastructure and dispenses the need for sorting, dilution or downcycling. By leveraging metallurgical principles and accelerated precipitation, the produced alloys achieve yield strengths that even surpass the commercial automotive alloy spectrum. This approach establishes a circular, low-emissions route to high-value aluminium recovery and offers a strategic model for transforming today´s and future´s critical raw material streams into next-generation structural alloys.

In 2017 alone, 7–9 million metric tons (MT) of end-of-life vehicle (ELV) scrap was generated across the European Union[1]. Recycling the associated materials presents substantial challenges, driven by a complex interplay of chemical, technological, economic and regulatory factors. From a thermodynamic perspective, removal of tramp elements introduced through aluminium (Al) scrap is inherently difficult, the more noble the tramp element[2–4]. They dissolve in the metal Al phase[2–4], demanding energy-intensive or technologically immature processes or environmentally hazardous reaction agents (e.g. $Cl_2$-gas-mix as purge gas) for a theoretical melt purification[5–7]. Current ELV recycling processes involve shredding and separation of steel and other materials from the Al fraction, but there are limits on the levels of purity reachable by these methods[8]. Strict regulatory alloy standards further complicate the production of high-quality wrought Al alloys via recycling[9]. An average European vehicle from 2019 contains 26 different Al alloys across 14 different applications[10,11]. In the conventional recycling process, depicted schematically in Fig. 1a (top left) below, the sorting of scrap is subject to natural limitations and often fails to completely separate the multiple Al alloys[8], also because they are joined (i.e. by welding[12]). Therefore, the production of wrought alloys, which must typically feature low amounts of alloying elements and are subject to strict chemical standards, requires substantial amounts of primary Al to dilute the concentration of tramp elements[8,13,14]. Only a comparably simple melt purification step targeting a specific, limited set of tramp elements (e.g. Na, Ca, Li, Mg) is proven operational[7]. Hence, the largest Al scrap stream by far is currently downcycled into cast alloys[14,15] for combustion engines, whose mechanical properties are greatly inferior to those of wrought alloys or primary structural cast alloys[9,16,17]. In 2012 alone, this stream amounted to 6.1 MT[18] globally. With declining demand for the alloys involved, it is projected that by 2030 no less than 6 MT of low-quality scrap will become unrecyclable each year[14,18]. This trend involves substantial economic costs, estimated at ~ 15 bn $ at 2025 Al prices[19], and will have severe

[1]Chair of Nonferrous Metallurgy, Montanuniversität Leoben, Franz-Josef Straße 18, 8700 Leoben, Austria. [2]Christian Doppler Laboratory for Deformation-Precipitation Interactions in Aluminum Alloys, Montanuniversität Leoben, Franz-Josef Straße 18, 8700 Leoben, Austria. [3]Christian Doppler Laboratory for Advanced Aluminum Alloys, Montanuniversität Leoben, Franz-Josef Straße 18, 8700 Leoben, Austria. ✉e-mail: stefan.pogatscher@unileoben.ac.at

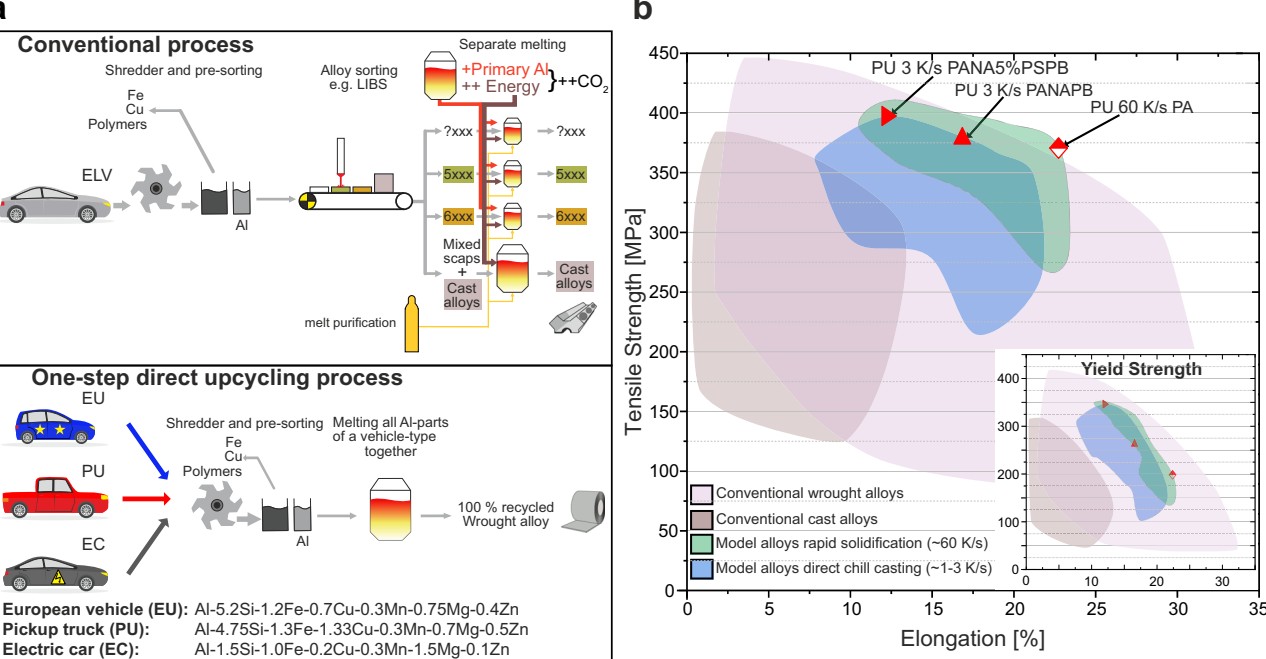

**Fig. 1 | Upcycled alloys from unsorted end-of-life vehicle (ELV) scrap.**
**a** Schematic comparison between the conventional production of individual alloy classes (top left) and direct aluminium-alloy upcycling from complete ELVs under industrial scalable (i.e. direct chill casting) and rapid solidification (i.e. thin-strip casting) conditions (bottom left). The ELV (upper box) and EC-icons and the crucible icons are reused and recoloured from our prior study (Krall, P., Weißensteiner, I. & Pogatscher, S. Recycling aluminium alloys for the automotive industry: Breaking the source-sink paradigm. Resources, Conservation and Recycling 202)[24]. This study is licensed under a CC BY 4.0 license. **b** Tensile strength (UTS)-ductility plot comparing today's commercial automotive alloys (conventional wrought and cast alloys) from Ansys® Granta Research Selector, Release 2025 R2, Level 3 Aero database[17] (Data reproduced courtesy of Ansys, Inc.) with alloys directly upcycled from complex ELV scrap mixtures from current average European vehicles (EU), electric cars (EC) and US pick-up trucks (PU). The diagram highlights the impact of casting conditions such as specifically the difference between rapid solidification (60 K/s) and widely used direct chill casting (1–3 K/s) on the mechanical performance of the upcycled alloys. The samples underwent various steps involving pre-aging (PA, 5 h at 100 °C), natural aging (NA, 14 d at 25 °C), paint-baking (PB, 20 min at 180 °C) and pre-straining (5%PS). A yield strength-ductility plot has been inserted into **b** using the same scale as that of the UTS plot.

environmental consequences, including an estimated[20,21] additional 90 MT of $CO_2$ emissions annually. Industry and academia must urgently address the current problem arising from the cars on the roads today in the coming years. However, designing sustainable cars and alloys[22,23] to prevent such new recycling dilemmas in the future is also crucial, but unfortunately seems far from global realisation.

Our one-step direct upcycling approach, presented schematically in Fig. 1a (lower left), addresses the current recycling challenge by bypassing the need for Al alloy sorting and eliminating the requirement to use primary Al to dilute impurities. Our approach disregards current alloy composition boundaries, especially those affecting silicon, iron and copper[13,24–27]. This contradicts the common rule in metal recycling. It posits that the purer the input materials the better their properties[18]. A reason for this behaviour is that many tramp elements form brittle intermetallic phases (IMPs) in Al[18].

However, in the context of steels Kim et al.[28]. showed that brittle IMPs can be utilised to increase strength at high ductility. Recently, we discussed a heterostructure approach[29] utilising brittle IMPs as the hard phase and Al as the soft phase to improve strain hardening in a 6xxx alloy[27]. We also assessed whether the concept could be considered for compositions generated by the Al alloys of various ELVs[24]. Our initial evaluation was conducted under rapid solidification conditions to control IMP morphology, conditions that are currently not applicable to the required industrial scale. Moreover, contamination with other metals from full ELVs recycling (i.e. Fe, Cu) were not considered, and the findings of that study[24] indicated modest yield strengths (150-190 MPa), below current 6xxx-series automotive standards. Further, many alloys must facilitate rapid strengthening upon the formation of nanoscale precipitates within the short heat treatment cycle used in manufacturing (called paint-baking, typically 20 min at 180 °C in the automotive

industry)[30], which was obviously not achieved in our initial study[24]. However, despite these shortcomings, the initial results motivated us to address these problems. While we also explored new processes (e.g. friction extrusion for solid-phase alloying[31]), we chose to use the well-established mass production metallurgical infrastructure, as only this will be able to address the global problem in time and scale.

In this study, we utilise the full Al-stream of ELV scrap under realistic industrial conditions, such as direct chill casting, which is the central aluminium casting technology in the aluminium industry[18], even more prominent for automotive products. We maximise grain refinement via particle-stimulated nucleation (PSN)[32], exploit heterostructure effects, and enhance mechanical properties via an accelerated precipitation-strengthening concept. This approach enables us to achieve an exceptional strength-ductility balance in directly upcycled alloys: yield strengths exceeded 350 MPa, and high ductility is maintained. Most importantly, the process aligns with prevailing mass-production conditions, ensuring its pressing integration into existing manufacturing infrastructure.

## Results and discussion
Compared to existing commercial automotive alloys, alloys directly upcycled from different vehicle types can reach an exceptional balance of strength and ductility (see Fig. 1b and Supplementary Fig. 2). The alloy derived from an average EU vehicle (EU)[10,11], representing the largest ELV scrap stream, has a markedly different composition than that of standardised alloys, as does the alloy from a US pickup truck (PU)[33,34]. In contrast, the electric-car-oriented alloy (EC)[35] shows elevated iron levels but remains compositionally closer to conventional 6xxx-series alloys. Not too surprisingly, the strength-ductility plot reveals that rapid solidification at a cooling rate of 60 K/s[36],

characteristic of some thin-strip casting techniques[37], delivers the most favourable basis for achieving good mechanical performance in further-processed sheets, consistent with previous findings[26,38]. What is fascinating, however, is that the performance of the directly upcycled alloy exceeds the tensile strength range (Fig. 1b) where current automotive wrought alloys (5xxx and 6xxx) are found. The key finding is that, under readily scalable industrial direct chill and continuous casting conditions (block-centre solidification rates of 1–3 K/s, higher at the ingot skins)[36,37,39,40], remarkable properties can also be retained. However, details are key. At 3 K/s the strength-ductility balance remains superior in most cases, whereas at 1 K/s mechanical properties can degrade significantly. Across all cooling rates, the mechanical properties of directly upcycled alloys outperform those of typical cast alloys, with only highly specialised (primary) cast alloys exhibiting comparable elongations[9,16,17].

As an example, we will examine the PU alloy during processing, as it demonstrates an exceptional strength-ductility balance while exhibiting significant sensitivity to processing conditions, such as a variation in achievable elongation at similar strength by a factor of 1.47 (NA) to 2.11 (2 % plastic pre-strain) (Supplementary Fig. 2). Figure 2 presents the mechanical and microstructural characteristics in both the as-cast and homogenised states, revealing a strong dependence on cooling rate. In the as-cast condition, elongation ranges from $0.5 \pm 0.3$ % at 1 K/s to $4.7 \pm 1.3$ % at 60 K/s, while in the homogenised state, it increases from $0.6 \pm 0.4$ % at 1 K/s to $11.3 \pm 0.6$ %. A similar trend is observed in the EU and EC alloys (Supplementary Fig. 3a, b). These results correlate with microstructural observations, which show clear refinement and morphological changes on the part of IMPs at an increasing cooling rate under both as-cast (Fig. 2b–d) and homogenised (Fig. 2e–g) conditions. Greater undercooling and lower temperatures generally promote nucleation and reduce diffusion rates[41,42]. IMP types containing Fe are known to shift from α-AlFeSi (Chinese-scripted) to β-AlFeSi (needles) at decreasing cooling rates, as these phases form inter-dendritically at lower cooling rates and superheating temperatures[26,43]. In light of IMP morphology and literature regarding cast alloys, β-AlFeSi phases are expected to have a more detrimental effect on ductility than rounded α-AlFeSi phases[26,43]. This is clear if 60 K/s is compared with 3 K/s, but less clear if we consider 1 K/s.

Upon homogenisation (Fig. 2e–g), the eutectic Si network undergoes spheroidisation through disintegration, spheroidisation and particle growth, leading to enhanced ductility[44,45]. IMPs containing Fe typically require annealing times of more than 150 h or temperatures above 560 °C to undergo significant modification[46], at which point the direct upcycling alloys are at risk of initial melting. Slow batch-cooling after homogenisation, in turn, triggers a transformation from α to β-AlFeSi[40]. The differences between 1 to 3 K/s are less obvious in the as-cast state than when 60 K/s is applied. However, after homogenisation for 3 K/s needle-like IMPs exist whose projected thickness ($< 3 \mu m$) is smaller than that of IMPs formed upon 1 K/s (projected thickness of $< 6 \mu m$). In the following, this is discussed as a parameter for improving ductility in deformed states, because thin needles are more advantageous than thicker spherical or cubic morphologies during fragmentation upon processing.

Figure 3a presents the mechanical performance of the PU alloy under selected processing and heat treatment conditions. Strain hardening, a key parameter for sheet formability[26,29,30], is significantly greater in the PU alloy, particularly in the PA condition, than that of the standard wrought automotive 6016 alloy. Faster cooling yielded higher strain-hardening potential; the EU alloy even exhibits a slightly higher strain-hardening potential, in contrast to the EC, which has a lower one. Additional data are provided in Supplementary Figs. 4c, 5c and 6c. Besides the obvious effect that an increasing solute content increases strain hardening[47–49], the superior strain hardening behaviour of EU and PU alloys originates primarily from the high volume fraction of IMPs in beneficial sizes (small) and morphology (aspect ratios of around 1)[26,50], as shown in Fig. 3b. This refined structure results from the initial structure after homogenisation and fragmentation upon rolling. In this morphology, IMPs enhance strain hardening via the formation of excess geometrically necessary dislocations (GNDs) (Fig. 3b–d), thereby increasing elongation through the concept of hetero-deformation-induced (HDI) strain hardening[29].

IMPs also promote PSN, thereby reducing Al grain size and improving formability. Notably, the average grain size (equivalent circle diameter) decreases with increasing cooling rate, from $21 \pm 7 \mu m$ at 1 K/s to $16 \pm 5.5 \mu m$ at 3 K/s, and further to $11 \pm 3.5 \mu m$ at 60 K/s. In regions surrounding IMPs, GNDs accumulate, creating favourable nucleation sites that facilitate grain refinement[26,27,32]. During recrystallisation, these sites lower the nucleation energy, while pinned grain boundaries suppress further grain growth[26,27,32]. This is corroborated by Kernel Average Misorientation (KAM) overlay maps of the strained samples (Fig. 3b–d),

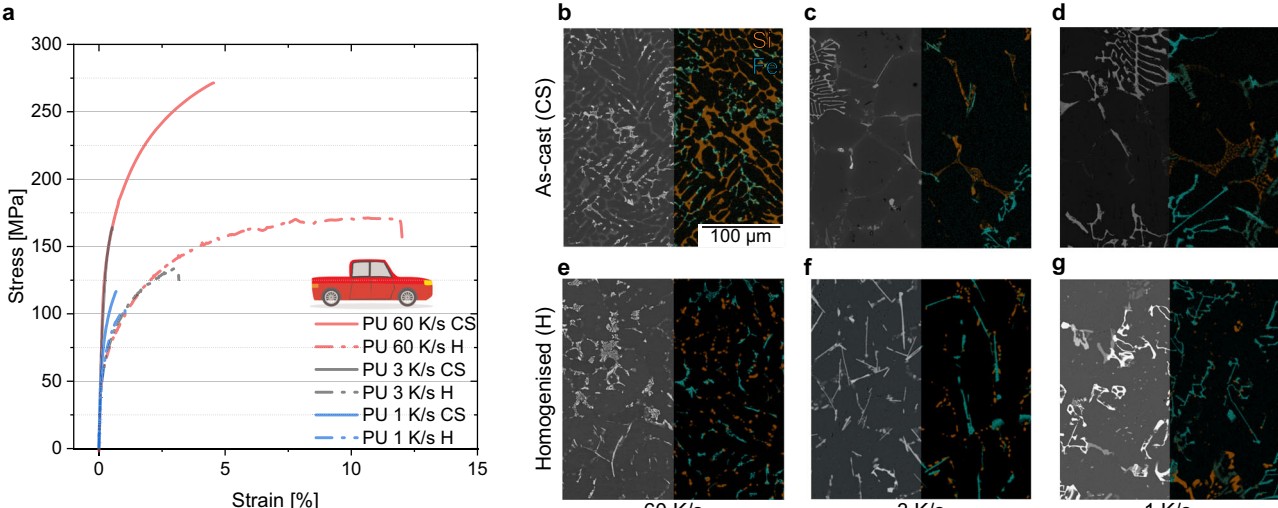

**Fig. 2 | Casting and homogenisation. a** Stress-strain curves for the pickup (PU) alloy across different cooling rates under as-cast (CS) and homogenised (H) conditions. **b–d** Energy dispersive X-ray scanning electron microscope (SEM/EDX) images of the PU alloy in the as-cast state, showing intermetallic particles (IMPs) containing Si and IMPs containing Fe and their coarsening at a decreasing cooling rate. **e–g** SEM/EDX images of the homogenised state, revealing the spheroidisation of IMPs containing Si and the transformation of IMPs containing Fe. The scale bar and colour key for Fe (blue) and Si (orange) in (**b**) is also valid for (**c–g**).

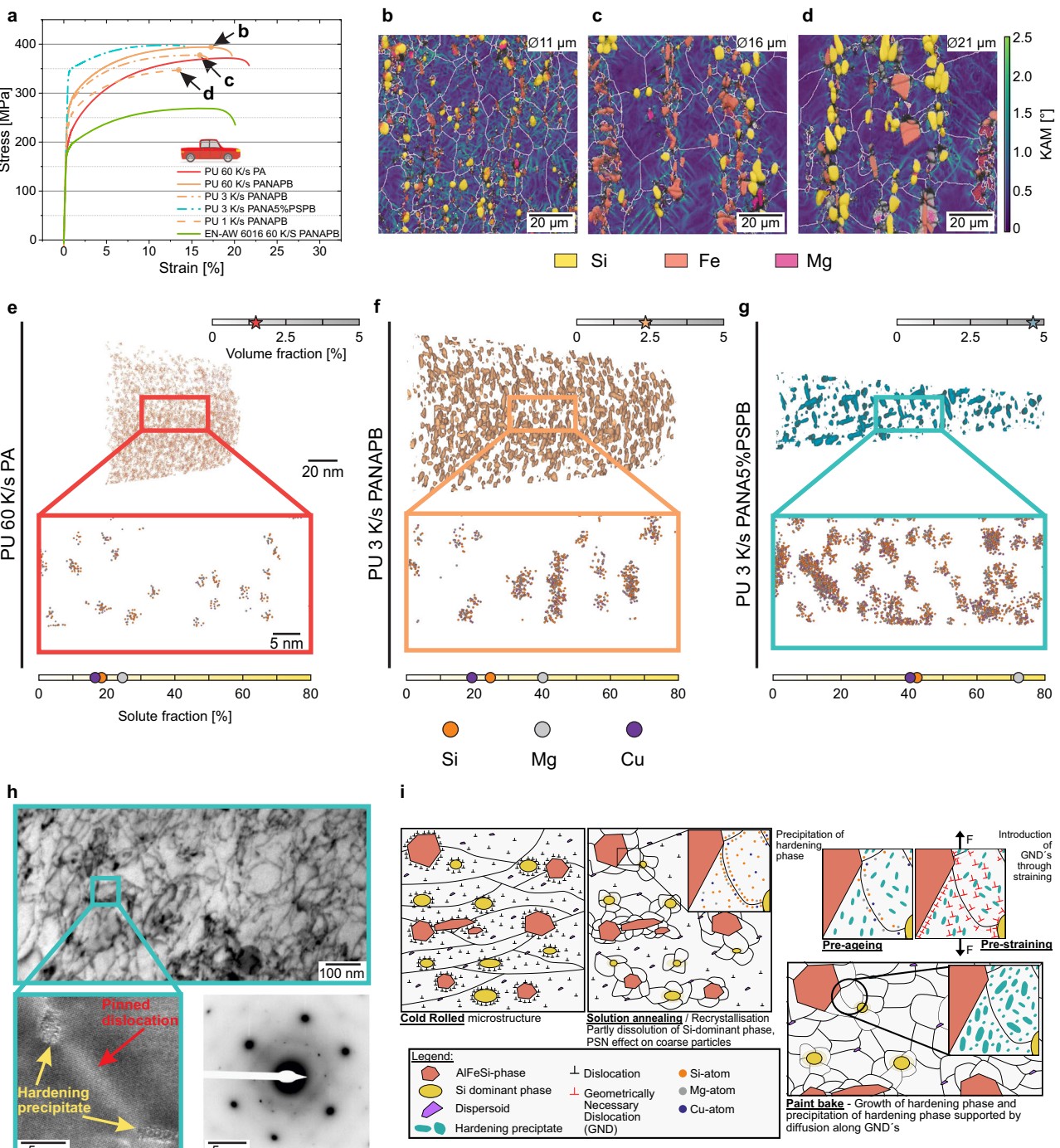

**Fig. 3 | Deformation and age-hardening performance in the rolled sheet material. a** Stress-strain curves of the pickup (PU) alloy under different processing and heat treatment conditions, compared to those of the automotive standard 6016 alloy. **b–d** Electron backscattered diffraction (EBSD)/Kernel average misorientation (KAM)/energy dispersive X-ray-overlay maps of the PU alloy after pre-aging, natural aging and paint-bake treatment (PANAPB), following different cooling rates upon casting measured at the uniform elongation. Grain boundaries are marked in white, with Si-dominant intermetallic particles (IMPs) in yellow, Fe-dominant IMPs in orange and Mg-dominant IMPs in pink. The mean grain size is indicated in the top right-hand corner. Structural refinement and geometrically necessary dislocation (GND) accumulation around IMPs enhance strain hardening and ductility.

Strengthening upon PB improves with pre-straining after natural aging (PANA5% PSPB, **a**). **e–g** Atom probe tomography data showing clusters/precipitates for PA, PANAPB and PANA5%PSPB with indication of the rising solute and volume (red, orange and teal stars) fraction with ongoing processing (Supplementary Table 3). The scale bars in **e** are valid for (**f** and **g**). **h** Transmission electron microscopy (TEM) image illustrating dislocation-precipitate interactions in PANA5%PSPB with high-resolution high-angle annular dark-field scanning-TEM insert and diffraction pattern to confirm the presence of Q′ hardening precipitates (see also Supplementary Fig. 9c). **i** Schematic representation of particle-stimulated nucleation, GND-driven strain hardening, and accelerated precipitation, which together enable direct upcycling alloys, surpassing the performance of commercial automotive alloys.

where higher misorientation values indicate a greater density of GNDs[27]. The KAM maps show that these elevated values are concentrated around IMPs, whereas the values at grain boundaries (marked by white lines) do not exceed the grain interiors significantly. Notably, this behaviour remains consistent regardless of cooling rate, pre-straining, or IMP chemistry (see Supplementary Information).

Refinement is most effective at 60 K/s, remains substantial at 3 K/s, and is least pronounced at 1 K/s. Changes in IMP characteristics (Figs. 2b–d and 3b–d) correlate directly with the observed decrease in ductility ($19 \pm 1.1$ % to $11 \pm 2.3$ %) for the PANAPB state as the cooling rate decreases (Fig. 3a).

In conclusion, the (unexpected) superior ductility of highly contaminated alloys arises from a synergistic effect of IMPs on grain refinement and enhanced strain hardening, both parameters being physically linked to increased uniform elongation[51]. This finding challenges conventional expectations of ductility in heavily contaminated alloys which until now have been limited to cast alloys. Currently, no wrought alloys exist within this complex compositional space that can leverage the beneficial role of IMPs while serving as sinks for tramp elements. However, the optimal type, size, and number density of IMPs remain undefined, leaving significant opportunities for further research.

An examination of the strengthening response of the PU alloy upon paint baking revealed a moderate value for PANAPB, which is slightly below the typical value of 6016[22]. However, adding pre-straining (PANA5%PSPB) paint baking generated a yield strength (YS) of $345 \pm 2$ MPa with $400 \pm 3$ MPa ultimate tensile strength (UTS) and $11 \pm 2$ % elongation at fracture (A) (Fig. 3a), a strengthening superior to that of automotive alloys. With values of $350 \pm 5$ MPa YS, $410 \pm 2$ MPa UTS and $12.3 \pm 0.9$ % A the 60 K/s samples range only a bit higher than the scalable direct chill casting variant at 3 K/s. The sample cooled at 1 K/s, however, shows lower strength and ductility (Supplementary Fig. 5a,b). The values for the EU alloy (Supplementary Fig. 4a, b) are only slightly lower than those of the PU alloy, although the composition differs (mainly 0.45 % more Si and 0.6 % less Cu), pointing at compositional robustness for internal combustion engine vehicles. The values of the EC range lower (Supplementary Fig. 6a, b).

To further assess microstructural properties on the atomistic scale, we show atom probe tomography (APT) data in Fig. 3e–g. Starting with small Si-Mg-Cu-clusters after PA (Fig. 3e), growth is observed when solely paint baking (Fig. 3f). If pre-straining is introduced, needle-shaped Q′ precipitates are formed during paint baking, which are known to have higher strengthening capability (Fig. 3g)[52]. Similar to 6xxx series alloys[25,53], the highest overall number density and smallest cluster size are observed in the PA condition, followed by the PANAPB material. Significantly, precipitates of 500 or more atoms or those larger than 5 nm appear only in pre-strained material (Supplementary Fig. 8a, b), indicating growth through pre-straining and paint-baking. The volume fraction of particles increases from 1.48 % in the PA condition to 2.34 % after PB and nearly doubles to 4.63 % after 5 % pre-straining. All of this supports precipitation enhancement by pre-straining[18,25,54]. This can be explained by the large fraction of additional excess vacancies and dislocations (see Supplementary Note 7 for the calculation and Supplementary Fig. 10), enhancing diffusion by a factor of $8.41 \cdot 10^5$ (for Mg) and even $2.35 \cdot 10^6$ for Cu. It is remarkable that the PANA5%PSPB route enables the material to achieve a structural state more comparable to a peak-aged condition rather than the underaged state typically associated with the short duration of a paint bake process, thereby resulting in the observed high yield strength[30].

Transmission electron microscopy (TEM) shows the increased dislocation density from PA material over the PANAPB state (see Supplementary Fig. 9a, b) towards the pre-strained sample (Fig. 3h), also indicating a higher number of dislocation-particle interactions. High-resolution TEM and STEM micrographs (inserts to Fig. 3h) show precipitate growth at pinned dislocations in the pre-strained material,

a behaviour not obvious in non-pre-deformed samples; this is also demonstrated schematically in Fig. 3i. The hardening precipitates are confirmed to be Q′ phase by the diffraction pattern and its reconstruction according to literature[55] (Supplementary Fig. 9c), which points to a more mature state then typically seen after a paint bake[30,52].

The results gathered from our mechanical and microscopic investigations indicate a dependence on the processing path. In the as-cast condition, the alloys exhibit brittleness and low elongation at fracture, which is significantly improved by homogenisation treatment. Subsequent rolling steps promote the distribution of IMPs, favourable for generating a fine final microstructure. By optimising the sequence of heat treatments and pre-straining steps following solution annealing and quenching, impressive strength and ductility were achieved under readily scalable direct chill casting conditions, opening a pathway to explore further properties (i.e. corrosion behaviour) for high-performance applications. While achieving this superior strength requires precise control over the final thermo-mechanical processing, the results also indicate that a wide range of properties is achievable (Supplementary Fig. 2).

In summary, this study effectively addresses the largest problem in future aluminium recycling and presents a simple, direct upcycling process for ELVs that produces wrought alloys without requiring scrap sorting or dilution with primary aluminium during melting. The high content of foreign elements in these alloys offers significant advantages which outweigh the potential downsides of slightly reduced elongations. To demonstrate the robustness of our approach, we addressed a range of alloy compositions and cooling conditions, including those achievable in industrial direct chill casting processes. We also explored various heat-treatment procedures and pre-straining parameters, producing combinations of mechanical properties that, surprisingly, even outperform typical automotive alloys in some cases. The comparable property range resulting from the average European car and US pickup alloys demonstrates notable compositional robustness. Achieving this robustness seems feasible with typical shredders processing more than 100 vehicles per hour[56] and typical recycling furnaces handling scrap from over 50 ELVs[57]. The sustainable process eliminates the need to (i) pre-sort different aluminium alloy groups, (ii) dilute with primary aluminium. Instead, it directly upcycles mixed aluminium scrap from ELVs into high-quality secondary wrought alloys. Crucially, our approach relies exclusively on a novel combination of processes already available at an industrial scale, offering a viable, immediately deployable pathway to overcome current downcycling and address the growing stream of nonferrous metal scrap.

## Methods

### Alloy production

The principal concept by which we calculated the composition of ELV alloys was presented in a previous study[24] (see Fig. 1, Supplementary Fig. 2 and Supplementary Table 1 and the underpinning literature data[9–11,33–35]). For comparison, the PU alloy was updated according to the approach of Zhu et al.[34] and the Fe level was increased by 0.7 % to simulate contamination by unremoved steel scrap. The alloys were synthesised in two different ways: melting on a laboratory scale (100–110 g batch size, $50 \times 42 \times 15$ mm³ ingot dimensions) in a miniaturised induction furnace (Indutherm, MC100V) and on a small scale ( ~ 8 kg batch size, $185 \times 85 \times 45$ mm³ "filet block" dimensions with representative cooling rate) in a resistance furnace (Nabertherm). To ensure high cooling speeds of ~60 K/s[36], representing thin-strip casting, a copper mould was deployed in the former. To represent direct chill casting cooling conditions, two pre-heated steel moulds were used. One of these was connected to Mettop's[58] ILTEC cooling system to raise the cooling rates from ~1 K/s (air cooling) to ~3 K/s[36]. Please note that this process was designed by our group in a previous work[36] aiming at producing industrially relevant

homogenous microstructures to represent the range of relevant cooling rates in DC casting in a technically feasible homogenous manner. Here, AlTi5B1 was also added to the melt for grain refinement at 0.1 m-%.

## Sample processing and mechanical characterisation

Nabertherm circulating air furnaces were deployed for the homogenisation process. This started with 10 h of heating from RT to 450 °C, after which the temperature was held for 10 h. After further heating at 10 K/s, the final temperature of 520 °C was held for 10 h, followed by furnace cooling.

For hot rolling, the ingots were milled to 12 mm in thickness for the laboratory scale procedure and to 18 mm for the small-scale procedure. They were then heated in the circulating air furnace to 430 °C within 1 h, and this temperature was held for 3 h before the first step. The ingots were hot rolled in a miniaturised rolling mill to 3.4 mm in thickness with step sizes of ~1 mm. Cold rolling was conducted from 3.4 mm to 2.4 mm with step sizes of 0.2 mm. Intermediate annealing in the circulated air furnace included heating to 370 °C within 4 h, holding this temperature for 10 h and furnace cooling. Final cold-rolling to 1.2 mm sheet thickness was again performed with step sizes of 0.2 mm. Standardised $A_{30}$ (30 mm initial gauge length) tensile samples were then milled from the finished sheets.

Solution annealing of tensile test samples was also conducted in the circulating air furnace, where a contact heater was placed. This ensured that the samples would reach the solution annealing temperature of 520 °C within one minute (mimicking industrial continuous heat treatment lines). The whole cycle took 10 min and was followed by subsequent water quenching.

The following heat treatment steps were carried out in circulating oil baths (Lauda): Pre-aging treatment, which took 5 h at 100 °C; and paint-baking for 20 min at 180 °C. Natural aging was performed in a Peltier-cooled incubator for 14 d at 25 °C.

For tensile tests under as-cast and homogenised conditions, for the small-scale ingots, 5 mm-thick slices were cut from the original ingot and milled to 2 mm in thickness and further to $A_{30}$ samples. For laboratory-scale ingots, however, rounded $A_{25}$ (25 mm initial gauge length) samples with threaded heads were manufactured.

The device used for the pre-straining and tensile testing was a Z100 from Zwick/Roell with a load cell of 50 kN and pneumatic sample holders. The vertical speed of the crosshead was 0.0025 s$^{-1}$ after $R_{p0.2}$; for a more accurate determination of $R_{p0.2}$ the speed was lowered to 0.00025 s$^{-1}$ around it. For pre-deformation, the machine was stopped at strains of 2 and 5 %. For a summary of the parameters see Supplementary Table 2.

## Microstructure analysis

**SEM/EBSD.** A 10 mm slice was cut off the ingots before and after homogenising. The slices were then ground and polished mechanically during the last step using an OPS agent.

Scanning electron microscopy (SEM) was conducted with an FE-SEM Jeol 7200 F instrument. Back-scattered-electron (BSE) micrographs were taken for the alloys investigated, and energy-dispersive X-ray (EDX) analysis of their six main alloying elements was performed. An Oxford Symmetry S2 detector was utilised for the EBSD. To enable a comparison between these different conditions, the results were determined as the equivalent circle diameter (ECD)[59]. Grain boundaries were defined as boundaries with a misorientation greater than 5 °. Al grains larger than 10 pixels were considered for grain size analysis. The kernel average misorientation was determined for the PU alloy.

**TEM.** For TEM investigations, samples of 20 × 10 mm were cut from the remaining sheets for analysis without pre-deformation. For pre-deformed samples, they were cut from the tensile samples. They were then ground to 80 μm in thickness, and 3 mm cylindrical samples

were stamped from the ground sheets. Electropolishing was then performed using a 25 % HNO$_3$-75 % methanol electrolyte as a polishing agent at a temperature range of −18 to −25 °C and voltages between 12 and 14 V. A ThermoFisher Scientific™ Talos F200X G2 machine was utilised at 200 kV to analyse the samples.

**APT.** For APT investigations, needle-shaped specimens (blanks) measuring 0.7 ×0.7 ×20 mm$^3$ were prepared from the head or gauge (mandatory for pre-strained samples) of a tensile sample after the final heat treatment. Further preparation involved a two-stage standard electropolishing process (1$^{st}$ stage: 25 % HNO$_3$ in methanol; 2$^{nd}$ stage: 2 % HClO$_4$ in 2-butoxyethanol) to minimise potential natural aging.

APT experiments were conducted in voltage mode with a pulse fraction of 20 %, a frequency of 200 kHz and a detection rate of 1 % at a temperature of 30 K using a LEAP 5000 XR. Data reconstruction[60] was performed using IVAS 3.8.14, with the maximum separation algorithm employed for cluster identification[61–63]. The 10$^{th}$-nearest-neighbour distance distribution method was deployed to determine the maximum separation of solutes within a cluster. The volumes of the APT reconstructions were quantified using the built-in "Alpha-Shape" function[64] in MATLAB™. The volume fraction was calculated as the ratio of cluster atoms to the total number of atoms within each APT specimen. The corresponding error margins for number densities and solute concentrations in matrix and clusters[65] were computed according to the methodology described in literature and the Guinier radii[66]. The composition of solute clusters was extracted from cluster search data using customised scripts[67], and the matrix composition by subtracting the number of atoms of solute elements (e.g., Mg, Si, Cu, Zn) in clusters and/or precipitates from the total number of atoms. For radial distribution function (RDF) analysis, Al, AlH1 and AlH2 represent Al. Random comparator curves are drawn around the existing atom positions 40 times within the set and the nearest-neighbour distribution[68,69]. The RDF can then be calculated according to Eq. (1), with $\mathbf{R}_i$ representing the position vector of a specified atom[68]:

$$RDF = \text{Hist}\left(||\mathbf{R}_i - \mathbf{R}_j||_2\right) i \neq j \tag{1}$$

## Data availability

The raw data generated in this study have been deposited in the Zenodo repository under the accession code Doi: 10.5281/zenodo.17192766[70].

## Code availability

The python script for evaluating the diffusion enhancement upon pre-straining is also deposited in the Zenodo repository under the accession code Doi: 10.5281/zenodo.17192766 [70].

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

## Acknowledgements
The work of P.K., P.D., M.A.T., T.K., S.S. and S.P. was funded/co-funded by the European Union (ERC, HETEROCIRCAL, 101124514, S.P.). Views and opinions expressed are, however, those of the author(s) only and do not necessarily reflect those of the European Union or the European Research Council. Neither the European Union nor the granting authority can be held responsible for them. The work of I.W., P.A. and B.T. was funded by the Christian Doppler Research Association within the framework of the Christian Doppler Laboratory for Deformation-Precipitation Interactions in Aluminium Alloys (I.W.) and the Christian Doppler Laboratory for Advanced Aluminium Alloys (S.P.). The financial support by the Austrian Federal Ministry of Labour and Economy, the National Foundation for Research, Technology and Development and the Christian Doppler Research Association is gratefully acknowledged. The research reported on here was supported by the Austrian Research Promotion Agency (FFG) in the context of projects 3DnanoAnalytics (FFG-No. 858040, S.P.) and Future Matter by APT (FFG-No. 884644, S.P.).

## Author contributions
P.K., P.A., I.W. and S.P. conceptualised the paper. P.K. was the leading research scientist, conducting material preparation, material processing, mechanical testing, sample preparation and writing the original draft. B.T. (EDX) and I.W. (EBSD and KAM) performed the SEM investigations. P.A. contributed to APT sample preparation, APT measurements and the interpretation of results. M.A.T. and T.K. performed the TEM measurements and interpreted the TEM results. P.D. wrote the Python script for the modelling of the yield strength. S.S. and P.K. illustrated the figures. S.P. supervised the research. All authors discussed and deliberated on the findings. All authors reviewed and approved the final version of the paper.

## Competing interests
The authors declare no competing interests.
