## [Transparent Peer Review file · Nature Communications]

Direct aluminium-alloy upcycling from entire end-of life vehicles

Corresponding Author: Professor Stefan Pogatscher

Version 0:

Reviewer comments:

Reviewer #1

(Remarks to the Author)

The study addresses the urgent challenge of recycling aluminum from end-of-life vehicles (ELVs). Current methods of recycling often result in the downgrading of mixed aluminum scrap into low-grade cast alloys, which requires sorting or dilution with primary aluminum. The authors propose a scalable process that upcycles mixed aluminum scrap from ELVs directly into high-performance alloys. This work is highly meaningful and impactful because it addresses challenges in aluminum sustainability and promotes a low-carbon pathway.

1. When discussing the issues of existing recycling methods, it would be comprehensive if the authors could mention not only the addition of primary aluminum to dilute but also the significant energy consumption and negative environmental impact of the chemical processes used to remove impurities. One example is the process of removing magnesium with chlorine or fluorine.

Update Figure 1a accordingly.

2. Considering the significant improvement in strength and sound ductility compared to all the PU samples with 6016 (Figure 3a), it would be nice to have an in-depth discussion here. Usually, the ductility of mixed recycled Al is poor because it contains a high content of tramp elements (e.g., Fe>1%), which exist as large IMPs and are detrimental to the ductility. But here, even the PU 1k/s sample shows good ductility after processing. Can authors explain more about how this method avoids the coarse IMP in conventional methods? Provide a microstructure made via conventional methods with the same chemical composition, if possible. I believe it would make the outstanding results of this work more prominent.

3. It is well known that the dimensions of castings affect the cooling rate, especially in the center of the casting. Could the author please specify the dimensions of the cast in this research work? Is there any difference/challenges when scaling up this method to the industry level? E.g., conventional cast and giga-cast?

4. Also, can authors provide the size of tensile specimens used in this study?

5. Some grammar mistakes and non-standard terminology

a. Rewrite the sentence from lines 30-32.

b. Rewrite the sentences from lines 57-59.

c. Suggest using "million metric tons (MT)" instead of "Mio. t"

d. Replace "für" with "for" at line 224

e. Suggest using "utilized/utilizing/specialized/homogenize" instead of "utilised/utilising/specialized/homogenise", etc.

f. I guess ND means "non-deformed", is that correct? Please spell out when using it for the first time.

Reviewer #2

(Remarks to the Author)

Thank you very much for submitting nice paper to Nature Communication. I have carefully read the manuscript. This paper contains very interesting technical concepts. Unfortunately, however, I have decided that it is appropriate to reject this paper due to the following reasons:

1) It does not state a clear conclusion as an academic paper, and is judged to merely restate the basic concepts presented in

their previous paper (Patrick Krall, Irmgard Weißensteiner, Stefan Pogatscher: Recycling aluminum alloys for the automotive industry: Breaking the source-sink paradigm, Resources, Conservation and Recycling, 202 (2024), 107370). If this paper had been submitted to this journal first, the decision might have been different; that is, this paper merely reaffirms the basic concepts of the previous paper and presents the results of some case studies.

2) The authors' discussion focuses on some of the mechanical properties of aluminum (tensile strength stress-strain property etc.). Actually they are showing very interesting results. On the other hand, the properties that practical metallic materials must satisfy are not limited to mechanical properties; physical and chemical properties (electric conductivity, corrosion resistance etc.) are also extremely important. These properties are highly dependent on chemical composition, so that it is essential to correlate the relationship between each property and composition. Such detailed information is best presented in a materials science journal rather than this journal.

3) The biggest problem in recycling aluminum is contamination by the uncontrollable alloying elements. This problem has been discussed by many researchers (Reuter, Nakajima, Graedel etc.) but has rarely been cited. This is a topic that is appropriate for the Nature Communication. The essential solution of this problem lies in controlling or removal of the alloying elements in secondary aluminum, which are present in much higher concentrations than in other base metal materials like steel. It must be said that the technology presented in this paper is merely a temporary solution (I understand the importance of a temporary solution). The secondary aluminum alloy developed by the authors will eventually be recycled again, which will undoubtedly result in further enrichment of alloying elements. The authors should provide a solution to this problem.

4) As a conclusion, the authors should be encouraged to revise and submit this paper to other higher materials science journal such as Acta Materialia, JALCOM, JOM etc. as the series papers.

Reviewer #3

(Remarks to the Author)

The authors describe an innovative approach for aluminium-alloy upcycling from entire end-of life (ELV) vehicles. The topic is definitely of interest for the international scientific community.

The proposed method looks extremely interesting and of high potential impact if compared with the methodologies actually in use both in industry and also mostly in research approaches.

The authors are requested to add some information and discussion regarding the robustness of the proposed methodology at the varying (usual scattering) of the composition scenarios and also for different heat treatment and processing sequences. At the moment such aspects look a bit implicit in the main document and should be better clarified.

With such modification the paper can be definitively considered for publication on the journal.

Version 1:

Reviewer comments:

Reviewer #1

(Remarks to the Author)

Thank you for addressing my comments on the manuscript. I am happy with the detailed responses provided by the authors.

Reviewer #3

(Remarks to the Author)

The required modifications were introduced in the paper. It can be accepted for publication.

Response to Referees

We would like to thank the reviewers for the valuable contributions that improved the quality of our manuscript. In the following, we address the reviewers' comments, to which we respond after careful discussion. The resulting changes in the manuscript are highlighted in **yellow**.

General revision:

The abstract was adapted to comply with the Nature Communications formatting instructions (150 words).

Reviewer #1 (Remarks to the Author):

The study addresses the urgent challenge of recycling aluminum from end-of-life vehicles (ELVs). Current methods of recycling often result in the downgrading of mixed aluminum scrap into low-grade cast alloys, which requires sorting or dilution with primary aluminum. The authors propose a scalable process that upcycles mixed aluminum scrap from ELVs directly into high-performance alloys. This work is highly meaningful and impactful because it addresses challenges in aluminum sustainability and promotes a low-carbon pathway.

Authors: We thank the reviewer for the appreciation of both the importance and the urgency of our work. We also thank for many constructive inputs that we address in the following points.

1. When discussing the issues of existing recycling methods, it would be comprehensive if the authors could mention not only the addition of primary aluminum to dilute but also the significant energy consumption and negative environmental impact of the chemical processes used to remove impurities. One example is the process of removing magnesium with chlorine or fluorine.

Update Figure 1a accordingly.

Authors: Thank you for this input. The text was improved to address this issue at two places:

Lines 32–36: (in combination with issue 5a) *They dissolve in the metal Al phase, demanding energy-intensive or technologically immature processes or environmental hazardous reaction agents (e.g. Cl₂-gas-mix as purge gas) for a theoretical melt purification⁵⁻⁷.*

Ref 7 was therefore added to the manuscript:

7 Friedrich, B. Mapping Study on Aluminium Melt Purification from Post Consumer Scrap. Available at https://international-aluminium.org/wp-content/uploads/2024/04/Mapping-Study_Full-Report_Final.pdf (2023).

Lines 47–49: *Only a comparably simple melt purification step targeting a specific limited tramp elements (e.g., Na, Ca, Li, Mg) is proven operational⁷.*

Fig 1a: In revised Fig. 1a, a yellow gas bottle was included to symbolise melt purification treatment. Also, a brown stream indicating energy input was added.

[Figure Redacted]

2. Considering the significant improvement in strength and sound ductility compared to all the PU samples with 6016 (Figure 3a), it would be nice to have an in-depth discussion here. Usually, the ductility of mixed recycled Al is poor because it contains a high content of tramp elements (e.g., Fe>1%), which exist as large IMPs and are detrimental to the ductility. But here, even the PU 1k/s sample shows good ductility after processing. Can authors explain more about how this method avoids the coarse IMP in conventional methods? Provide a microstructure made via conventional methods with the same chemical composition, if possible. I believe it would make the outstanding results of this work more prominent.

Authors: Thank you for this question. We will further discuss this discovery we report on. We performed the following changes in the manuscript:

The sentence “*Unlike Si-dominant IMPs spheroidised during homogenisation, Fe-dominant IMPs are primarily refined through rolling*” is moved from line 198 in the original manuscript to line 176 in the revised manuscript in order to clarify the discussion in the paragraphs from line 181 to 212, which extensively deals with the influence of IMPs on the final properties.

In the as-cast condition, the ductility is very poor for 1 and 3 K/s. This is also what the community often expects for impurities in cast alloys. Important is also that only cast alloys – for example – could accept this large amount of tramp elements: simply no wrought alloys exist in this compositional complex space. **Nevertheless, when Fe-dominant IMPs are**

morphologically modified (small and nearly spherical) and evenly distributed in the matrix via rolling, they deliver a substantial contribution to strain hardening, and consequently increasing the uniform elongation. They also play a key role for the particle stimulated nucleation upon recrystallising. In the surrounding of the IMPs, geometrically necessary dislocations are stored and are beneficial sites for the nucleation of new grains.

Furthermore, these particles act as obstacle for grain boundaries and therefore, the overall microstructure is grain-refined (sizes between 11–21 μm in our work) than in conventional 6016 alloys (average sizes between 22–32 μm) with a low number density of IMPs. Both increased strain hardening and grain refinement mitigate the negative impact of the high-impurity content. However, coarse sludge phases formed during casting however are obligatory to avoid. Emerging from our research paper, the optimum combination between IMP sizes and number density is still not defined, stimulating further research in this emerging field of research.

We summarize this in a concluding remark on the ductility behaviour shown in Fig. 3 (see line 214 to 221): *In conclusion, the (unexpected) superior ductility of highly contaminated alloys arises from a synergistic effect of IMPs on grain refinement and enhanced strain hardening, both parameters being physically linked to increased uniform elongation⁵¹. This new finding challenges conventional expectations of ductility in heavily contaminated alloys – which until now – have been limited to cast alloys. Currently, no wrought alloys exist within this complex compositional space that can leverage the beneficial role of IMPs while serving as sinks for tramp elements. However, the optimal type, size, and number density of IMPs remain undefined, leaving significant opportunities for further research.*

51. Dieter, G. E. Mechanical metallurgy. 2nd ed. (McGraw-Hill, New York, 1976).

3. It is well known that the dimensions of castings affect the cooling rate, especially in the center of the casting. Could the author please specify the dimensions of the cast in this research work? Is there any difference/challenges when scaling up this method to the industry level? E.g., conventional cast and giga-cast?

Authors: We referred to our previous work on representative sample production in the methods section, but without mentioning details that this was developed to represent DC casting. We now added to the methods section alloy production. To clarify, we made the following changes:

We added in the dimensions of the ingots for 60 K/s (*50x42x15 mm³ ingot dimensions*) in lines 322–323 and specified for 1 and 3 K/s (*185x85x45 mm³ “filet block” dimensions with representative cooling rate*) in line 324–325 and in lines 329 to 332: *Please note that this process was designed by our group in a previous work³⁶ aiming at producing industrial relevant*

homogenous microstructures to represent the of relevant cooling rates in DC casting in a technical feasible homogenous manner.

To address the upscaling question: We added “*industrial*” in line 110, as direct chill casting represents the major industrial casting process. We also added “*and continuous*” casting in line 111, as it offers the same cooling speeds as direct chill casting. **These facts mean that we already recreated the industrial cooling conditions in this research.**

A casting block is never homogeneous. We therefore specified in lines 111 and 112 from (1–3 K/s) to (*block centre solidification rates of 1–3 K/s, higher at the ingot skins*)^{36,37,39,40}. We also added reference 39, which give an insight on solidification and further inhomogeneous cooling speeds across the cross section of the castings. Giga-castings also show big inhomogeneities. However, as giga-castings are the finished product, no further processing – e.g. rolling – is performed in their fabrication.

39. Granger, D. A. *Microstructure Control in Ingots of Aluminum Alloys with an Emphasis on Grain Refinement. In Essential Readings in Light Metals, edited by J. F. Grandfield & D. G. Eskin (Springer International Publishing, Cham, 2016), pp. 354–365.*

4. Also, can authors provide the size of tensile specimens used in this study?

Authors: An explanation of A_{30} (30 mm initial gauge length) and A_{25} (25 mm initial gauge length) is added in lines 349 and 364–365.

5. Some grammar mistakes and non-standard terminology

Authors: We thank the reviewer for highlighting these issues, we will address them in detail point to point:

a. Rewrite the sentence from lines 30-32.

Authors: This sentence is split now as follows (lines 32–36):

From a thermodynamic perspective, removal of tramp elements introduced through aluminium (Al) scrap is inherently difficult the more noble the tramp element²⁻⁴. They dissolve in the metal Al phase²⁻⁴, demanding energy-intensive or technologically immature processes or environmentally hazardous reaction agents (e.g., Cl₂-gas-mix as purge gas) for a theoretical melt purification⁵⁻⁷.

We also added references 2–4 and 7 to fortify the sentences:

2. Reuter, M. A., van Schaik, A., Ignatenko, O. & Haan, G. J. de. *Fundamental limits for the recycling of end-of-life vehicles. Minerals Engineering* **19**, 433–449; 10.1016/j.mineng.2005.08.014 (2006).
3. Nakajima, K. et al. *Thermodynamic analysis of contamination by alloying elements in aluminum recycling. Environmental Science & Technology* **44**, 5594–5600; 10.1021/es9038769 (2010).
4. Graedel, T. E. et al. *What Do We Know About Metal Recycling Rates? Journal of Industrial Ecology* **15**, 355–366; 10.1111/j.1530-9290.2011.00342.x (2011).

b. Rewrite the sentences from lines 57-59.

Authors: Thank you, we split the sentences as follows (lines 66-68)

This contradicts the common rule in metal recycling. It posits that the purer the input materials the better their properties. A reason for this behaviour is that many tramp elements form brittle intermetallic phases (IMPs) in Al¹⁸.

c. Suggest using “million metric tons (MT)” instead of “Mio. t”

Authors: We changed the unit as suggested in the lines 29, 51, 52, 55.

d. Replace “für” with “for” at line 224

Authors: Thank you, we replaced it.

e. Suggest using “utilized/utilizing/specialized/homogenize” instead of “utilised/utilising/specialized/homogenise”, etc.

Authors: We unified the spelling to British English (utilising etc.), as in the title the word “aluminium” instead of “aluminum” is used.

f. I guess ND means “non-deformed”, is that correct? Please spell out when using it for the first time.

Authors: The acronym ND was used in a former version; we removed ND from the caption of Fig. 2 and in line 385.

Reviewer #2 (Remarks to the Author):

Thank you very much for submitting nice paper to Nature Communication. I have carefully read the manuscript. This paper contains very interesting technical concepts. Unfortunately, however, I have decided that it is appropriate to reject this paper due to the following reasons:

Authors: We thank the reviewer for their opinion on our paper. In the following, we have addressed all comments in this following response. In the points where we disagree with the reviewer, appropriate modifications have been made into the paper, increasing its scientific quality and making the discussion of the new results even more robust. Thank you.

1) **It does not state a clear conclusion as an academic paper**, and is judged to merely restate the basic concepts presented in their previous paper (Patrick Krall, Irmgard Weißensteiner, Stefan Pogatscher: Recycling aluminum alloys for the automotive industry: Breaking the source-sink paradigm, Resources, Conservation and Recycling, 202 (2024), 107370). If this paper had been submitted to this journal first, the decision might have been different; that is, this paper merely reaffirms the basic concepts of the previous paper and presents the results of some case studies.

Authors: We respectfully disagree with the assertion that our paper lacks a clear academic conclusion. The last two paragraphs of our manuscript draft show the conclusions of our paper, which we again summarise herein for the reviewer:

- Homogenisation and rolling refine structure, boosting ductility and strength.
- DC casting conditions enable scalable production of high-performance alloys.
- Mixed ELV scrap can be directly upcycled without sorting or primary dilution.
- Optimised heat treatment and pre-straining deliver superior mechanical properties. (we also emphasized this in an additional comment in line 292 to 294)
- Resulting alloys match or surpass automotive grades, showing strong robustness.

Moreover, our present research differs significantly from the previous work cited (Krall et al., 2024). For instance, in present study, we addressed on industrially relevant cooling rates typical of DC casting, rather than focusing solely on rapid solidification as in the previous paper. This makes our new recycling approach directly applicable to industrial processes. Additionally, we consider contamination arising from the shredding process (e.g., Fe impurities), rather than theoretical mixtures of pure aluminum alloys.

A key advancement presented in our paper is the development of a thermomechanical processing route that enables rapid hardening under industrial conditions (e.g., 20 minutes during the paint bake process). This results in yield strength values that are double those reported in the previous study. This aspect we also emphasize in an amended version of the manuscript (line 244, line 268 to 272 and line 281 to 282). Moreover, we provide a detailed understanding of nano-scale structure evolution via TEM, APT and through kinetic analysis (given in detail in the SI). While the paper builds upon the expertise acquired in our previous works – as is the case with any scientific publication – the specific aspects addressed here, such as, industrially relevant conditions, contamination levels, the development of a rapid hardening, and the mechanistic insights were never previously covered.

2) The authors' discussion focuses on some of the mechanical properties of aluminum (tensile strength stress-strain property etc.). Actually they are showing very interesting results. On the other hand, the properties that practical metallic materials must satisfy are not limited to mechanical properties; physical and chemical properties (electric conductivity, corrosion resistance etc.) are also extremely important. These properties are highly dependent on chemical composition, so that it is essential to correlate the relationship between each property and composition. Such detailed information is best presented in a materials science journal rather than this journal.

Authors: We understand the reviewer's comment regarding the importance of physical and chemical properties, in addition to mechanical properties. In our study, we focused on the mechanical properties, as these are the most critical for structural applications, particularly in automotive manufacturing. Specifically, properties such as strength, elongation, and strain hardening are essential for processes like deep drawing and stretch forming. We have demonstrated in our research that the strain hardening behaviour of our upscaled alloys is excellent. While we acknowledge the significance of properties such as corrosion resistance and electrical conductivity, these are often more application-specific. For example, corrosion resistance is less critical for inner car body components, and mitigation strategies (i.e. coatings, claddings etc.) are already in place to be further studied to expand the field of potential applications. To address this, we have added a comment in line 291 noting that corrosion behavior will be explored in future studies. Similarly, while electrical conductivity is important in certain contexts, it is less relevant for structural car parts, and there are known methods to enhance it when necessary (e.g. Boron melt treatment or overaging). We agree that a comprehensive investigation of the relationships between chemical composition and further specific properties is valuable. However, we believe that such detailed studies are better suited for materials science journals, as the reviewer suggests. Our current work represents a

foundational step, and we view this exploration as a natural progression for future research in both academia and industry.

3) The biggest problem in recycling aluminum is contamination by the uncontrollable alloying elements. This problem has been discussed by many researchers (Reuter, Nakajima, Graedel etc.), but has rarely been cited. This is a topic that is appropriate for the Nature Communication.

Thank you for recognising that our research topic is appropriate for Nature Communications.

The essential solution of this problem lies in controlling or removal of the alloying elements in secondary aluminum, which are present in much higher concentrations than in other base metal materials like steel. It must be said that the technology presented in this paper is merely a temporary solution (I understand the importance of a temporary solution). The secondary aluminum alloy developed by the authors will eventually be recycled again, which will undoubtedly result in further enrichment of alloying elements. The authors should provide a solution to this problem.

Authors: We thank the reviewer for highlighting this critical issue in aluminium recycling. We have now cited the relevant works by Reuter, Nakajima, and Graedel (References 2–4, line 34) to acknowledge the broader context of this challenge. **We recognise that the research herein represents a temporary solution, our paper is timely considering the state-of-the-art. Given the urgency of this issue, we believe that our present solution address the recycling challenges for the upcoming decades (~20 years).** The concept presented in our work aims at closing the recycling loop for Al alloys. As the use of Al in passenger vehicles continues to grow, driven by high-purity wrought and cast alloys, the compositions we propose represent an upper limit for vehicle compositions. Over time, these compositions are expected to dilute, as a 100% recycling rate is impossible to be achieved due to economic and energetic constraints. Additionally, the increasing demand for Al will further mitigating the enrichment of alloying elements in the next years to come. We fully agree with the reviewer that, beyond addressing this current recycling problem, it is essential to prevent such issues from arising again in the future. To this end, we are actively working on solutions for the automotive industry to narrow down the number of alloys in use and reduce compositional complexity by increasing their property range. For example, our work on crossover alloys (Ref: 22: On the potential of aluminum crossover alloys. Progress in Materials Science 124, 100873; and Ref: 18: Making sustainable aluminum by recycling scrap: The science of “dirty” alloys. Progress in Materials Science 128, 100947), aims to address this issue improving future ELV recyclability. We now address in line 56 to 59 of the manuscript:

Industry and academia must urgently address the current problem arising from the cars on the roads today in the coming years. However, designing sustainable cars and alloys^{22,23} to prevent such new recycling dilemmas in the future is also crucial, but unfortunately seems far from global realisation.

23. Gaustad, G., Olivetti, E. & Kirchain, R. *Design for Recycling. Journal of Industrial Ecology* **14**, 286–308; 10.1111/j.1530-9290.2010.00229.x (2010).

4) As a conclusion, the authors should be encouraged to revise and submit this paper to other higher materials science journal such as Acta Materialia, JALCOM, JOM etc. as the series papers.

Authors: We respectfully disagree with the reviewer's suggestion. While we acknowledge the value of publishing in specialized materials science journals, the goal of this work is to reach a broader audience, including policymakers, industry stakeholders, and society at large, to drive the paradigm shift urgently needed in this field. We believe that this journal provides the appropriate platform to engage with this wider community.

Reviewer #3 (Remarks to the Author):

The authors describe an innovative approach for aluminium-alloy upcycling from entire end-of life (ELV) vehicles. The topic is definitively of interest for the international scientific community.

The proposed method looks extremely interesting and of high potential impact if compared with the methodologies actually in use both in industry and also mostly in research approaches.

Authors: We thank the reviewer for the appreciation of the high potential impact of our work.

The authors are requested to add some information and discussion regarding the robustness of the proposed methodology at the varying (usual scattering) of the composition scenarios and also for different heat treatment and processing sequences. At the moment such aspects look a bit implicit in the main document and should be better clarified.

Authors: We thank the reviewer for their comment. We now address robustness of properties and handling of compositional scatter.

We amended/added the following text passages in the revised manuscript:

Revised manuscript lines 249 to 253: The values for the EU alloy (Supplementary Figure 4a,b) are only slightly lower than those of the PU alloy, although the composition differs (mainly 0.45 % more Si and 0.6 % less Cu) pointing at compositional robustness for internal combustion engine vehicles.

Revised manuscript lines 292 to 294: While achieving this superior strength requires precise control over the final thermo-mechanical processing, the results also indicate that a wide range of properties is achievable (Supplementary Figure 2).

Revised manuscript lines 300–302: To demonstrate the robustness of our approach we addressed a range of alloy compositions and cooling conditions, including those achievable in industrial direct chill casting processes.

In lines 305 to 309, we added a statement on the compositional robustness and related it to industrial practise and also added references 56 and 57:

The comparable property range resulting from the average European car and US pickup alloys demonstrate notable compositional robustness. Achieving this robustness seems feasible with typical shredders processing more than 100 vehicles per hour⁵⁶ and typical recycling furnaces handling scrap from over 50 ELVs⁵⁷.

56. Sander, S., Schubert, G. & Jäckel, H.-G. The fundamentals of the comminution of metals in shredders of the swing-hammer type. International Journal of Mineral Processing 74, S385-S393; 10.1016/j.minpro.2004.07.038 (2004).

57. *Krone, K. Aluminiumrecycling. Vom Vorstoff bis zur fertigen Legierung (Vereinigung Deutscher Schmelzhütten, Düsseldorf, 2000)*

With such modification the paper can be definitively considered for publication on the journal.

Authors: We thank the reviewer for this suggestion.